# Deciphering Transcriptional Networks during Human Cardiac Development

**DOI:** 10.3390/cells11233915

**Published:** 2022-12-03

**Authors:** Robin Canac, Bastien Cimarosti, Aurore Girardeau, Virginie Forest, Pierre Olchesqui, Jeremie Poschmann, Richard Redon, Patricia Lemarchand, Nathalie Gaborit, Guillaume Lamirault

**Affiliations:** 1Nantes Université, CHU Nantes, CNRS, INSERM, l’institut du thorax, F-44000 Nantes, France; 2INSERM, Nantes Université, Center for Research in Transplantation and Translational Immunology, UMR 1064, ITUN, F-44000 Nantes, France

**Keywords:** stem cell differentiation, human induced pluripotent stem cells, heart development, transcription factor, gene regulatory networks, transcriptomics, transcription factor complexes, iroquois transcription factors

## Abstract

Human heart development is governed by transcription factor (TF) networks controlling dynamic and temporal gene expression alterations. Therefore, to comprehensively characterize these transcriptional regulations, day-to-day transcriptomic profiles were generated throughout the directed cardiac differentiation, starting from three distinct human- induced pluripotent stem cell lines from healthy donors (32 days). We applied an expression-based correlation score to the chronological expression profiles of the TF genes, and clustered them into 12 sequential gene expression waves. We then identified a regulatory network of more than 23,000 activation and inhibition links between 216 TFs. Within this network, we observed previously unknown inferred transcriptional activations linking IRX3 and IRX5 TFs to three master cardiac TFs: GATA4, NKX2-5 and TBX5. Luciferase and co-immunoprecipitation assays demonstrated that these five TFs could (1) activate each other’s expression; (2) interact physically as multiprotein complexes; and (3) together, finely regulate the expression of *SCN5A*, encoding the major cardiac sodium channel. Altogether, these results unveiled thousands of interactions between TFs, generating multiple robust hypotheses governing human cardiac development.

## 1. Introduction

Heart formation is a complex process that requires spatio-temporal interplay between distinct and interdependent cell types through specific signaling and transcriptional pathways, leading to their differentiation and specification [1,2]. Defects in this developmental process result in congenital heart disease as well as in a number of inherited cardiac disorders in adults [3]. The specific gene expression program governing the formation of a functional heart needs precise regulation in a time-, cell-, and space-dependent manner [4]. This program is mediated by transcription factors (TFs) regulating the expression of other TF-encoding genes and establishing specific TF networks, such as between GATA4, NKX2-5 and TBX5 [5,6]. Over time, these networks control and permanently remodel the transcriptional expression program that govern heart development.

A thorough understanding of these networks is crucial to gain knowledge on the transcriptional regulations and dysregulations that govern normal and pathological cardiac development, respectively. However, complete knowledge of the global TF regulatory network of cardiac development is still missing. For instance, while several studies on Iroquois homeobox TF family (IRX) have shown their key roles on the regulation of adult cardiac electrical conduction [7,8,9,10,11], their function during human cardiac development has not yet been investigated. Cellular models derived from human induced Pluripotent Stem Cells (hiPSCs) offer a unique opportunity to address these challenges as they reproduce the cellular differentiation processes which lead stem cells to acquire a cardiac cell phenotype, carrying the genome of either healthy subjects or patients with inherited cardiac diseases.

In the present study, we first validated the hiPSC cardiac differentiation model as a relevant tool to unravel the global TF regulatory network governing human cardiac development, identifying a network of 216 TFs with time-dependent activations and inactivations. Among these, we identified and biologically validated an undescribed TF regulatory network involving IRX3, IRX5 and three main cardiac TFs: GATA4, NKX2-5 and TBX5. Furthermore, we generated new hypotheses on the potential mechanisms leading to the cooperative effect of these TFs that could form a functional multiprotein complex activating the promoter of *SCN5A*, encoding the main cardiac sodium channel.

## 2. Materials and Methods

### 2.1. Reprogramming and Maintenance of hiPSCs

All of the cell lines, taken from 3 healthy donors, were previously characterized. The hiPSC-A (C2a in [12]) line was generated using the lentivirus method while the hiPSC-B (IRX5-Wt in [13]; RRID:CVCL_B5QD) and hiPSC-C (WT8288 in [14]; RRID:CVCL_B5Q5) lines were generated using the Sendai virus method. The hiPSC lines were maintained at 37 °C, 5% CO_2_, 21% O_2_ in StemMACS^TM^ iPS Brew XF Medium (Miltenyi Biotec, Bergisch Gladbach, Germany) on culture plates coated with Matrigel^®^ hESC-Qualified Matrix (0.05 mg/mL, Corning, NY, USA). At 75% confluency, the cells were passaged using Gentle Cell Dissociation Reagent (STEMCELL^TM^ Technologies, Vancouver, Canada).

### 2.2. Cardiac Differentiation of hiPSCs

Directed cardiac differentiations of the hiPSCs were performed using the established matrix sandwich method (Figure 1A; [15]). When the hiPSCs reached 90% confluency, an overlay of Growth Factor Reduced Matrigel (0.033 mg/mL, Corning) was added. Differentiation was initiated 24 h later by culturing the cells in RPMI1640 medium (Thermo Fisher Scientific, Waltham, MA, USA) supplemented with B27 (without insulin, Thermo Fisher Scientific), 2 mM L-glutamine (Thermo Fisher Scientific), 1% NEAA (Thermo Fisher Scientific), 100 ng/mL Activin A (Miltenyi Biotec), 1X Pen/Strep (Thermo Fisher Scientific) and 10 ng/mL FGF2 for 24 h. Subsequently, on the next day, the medium was replaced by RPMI1640 medium supplemented with B27 without insulin, 2 mM L-glutamine, 1% NEAA, 10 ng/mL BMP4 (Miltenyi Biotec), 1X Pen/Strep and 5 ng/mL FGF2 for 4 days. By day 5, cells were cultured in RPMI1640 medium supplemented with B27 complete (Thermo Fisher Scientific), 2 mM L-glutamine, 1X Pen/Strep and 1% NEAA and changed every two days until day 30. Specifically, for video analysis and immunofluorescence staining, glucose starvation was performed to obtain a purified cardiomyocyte population: at day 10 the medium was replaced by Depletion medium (RPMI 1640 medium without glucose (Thermo Fisher Scientific) supplemented with B27 complete, and 1X Pen/Strep) for 3 days. The cells were dissociated at day 13 with 10X TrypLE solution (Thermo Fisher Scientific) and replated in CMs medium (RPMI1640 medium supplemented with B27 complete, 2 mM L-glutamine, 1X Pen/Strep, 1% NEAA) supplemented with Y-27632 Rho-kinase inhibitor (STEMCELL^TM^ Technologies). On day 14, the medium was replaced by Depletion medium for 3 days. From day 17, the cells were maintained in a CMs medium.

### 2.3. Bulk Transcriptomics

#### 2.3.1. RNA Extraction and Sequencing

For each hiPSC line, the samples were harvested daily, from D-1 to D30 of the cardiac differentiation protocol, from three independent cardiac differentiations. The total RNA were extracted using the NucleoSpin RNA kit (MACHEREY-NAGEL, Hoerdt, France) and their quality was assessed by NanoDrop^TM^ 1000 Spectrophotometer (Thermo Fisher Scientific). From the D-1 to D14 samples, all of the cells were collected while, from D15 to D30, to obtain samples enriched with cardiomyocytes, only the spontaneously beating cell clusters were collected, following mechanical isolation using a needle. Three RNA libraries were prepared by GenoBiRD core facility according to their published method [16] and sequenced on 8 individual runs on a NovaSeq 6000 or HiSeq 2500 Sequencing System (Illumina, San Diego, CA, USA).

#### 2.3.2. Primary Analysis of Bulk Transcriptomic Data

Demultiplexing, alignment on the GRCh38 reference genome and counting steps were conducted on each sequencing run with the Snakemake pipeline developed by the GenoBiRD core facility [16]. Normalized and log-transformed expression matrices were generated using the multiplates function correcting potential batch effects by treating cardiac differentiation time points as replicates.

#### 2.3.3. PCA

Principal Component Analysis (PCA) was performed with the R package FactoMineR ([17]; RRID:SCR_014602) on the entire mean-centered and log-transformed matrix.

#### 2.3.4. Time-Course Gene Expression Analysis

Genes with significant expression variation between the different cardiac differentiation time-points (indicated as Differentially Expressed Genes; DEG) were identified by multivariate empirical Bayes statistics using the R package timecourse ([18]; RRID:SCR_000077) and applied to the entire log-transformed matrix. We selected the top 3000 DEG based on their highest Hotelling ${\tilde{T}}^{2}$ statistics. The same method was used to select genes with significant expression variation during murine cardiac development from a published transcriptomic dataset [19]. When necessary, human and murine orthologous gene names were identified using the R package biomaRt ([20]; RRID:SCR_019214) and Ensembl databases.

#### 2.3.5. Clustering and Heatmap

The DEG were grouped into clusters, based on their expression level variation across the 288 samples, using the R function k-means set on 2000 iterations, and visualized with the R package ComplexHeatmap ([21]; RRID:SCR_017270).

#### 2.3.6. Gene Ontology Analyses

Gene Ontology (GO) analysis was performed using the R package ClusterProfiler ([22]; RRID:SCR_016884), based on GO Biological Process terms from the org.Hs.eg.db_3.14.0 and org.Mm.eg.db_3.14.0 databases for human and mouse annotations, as appropriate. Significantly enriched (bonferroni-corrected *p*-value < 0.05) biological processes, as compared to reference transcriptome, and with a Gene Set Size (GSSize) between 10 and 500, were considered for further analysis. The 15 GO terms with the lowest corrected *p*-value were visualized with treeplot.

#### 2.3.7. Network Construction and Analysis

For each hiPSC line, the gene regulatory network was inferred using the R package LEAP (Lag-based Expression Association for Pseudotime-series; Specht and Li, 2017), based on the average from the log-transform data of triplicate cardiac differentiations. Cardiac differentiation time points were used to rank samples, as required, by the LEAP tool. The max_lag_prop parameter was set to 1/10, meaning that, at most, 3 day windows were used to calculate the maximum absolute correlation (MAC) score. Only links with a significant MAC score (determined by a permutation test; *p*-value < 0.05) and related to a non-null time delay were considered. Links with a positive correlation score were interpreted as activation relationships and those with a negative correlation score as repression relationships. STRING software [23] was used to obtain information on the physical and functional interactions between the proteins of interest, with a minimum required interaction score of 0.4. Nodes without any interaction were excluded. STRING-based or LEAP-based interactions were processed using Cytoscape 3.9.1 for network reconstruction ([24]; RRID:SCR_003032). Network parameters were obtained using the Analyze network function.

### 2.4. Single-Cell Transcriptomic

#### 2.4.1. Single-Cell RNA-Seq Data Generation

Cells at D30 of hiPSC-A cardiac differentiation were harvested from three distinct beating wells, dissociated, using the Multi Tissue Dissociation Kit 3 (Miltenyi Biotec), and pooled. This experiment was performed in duplicates. Cell suspensions were filtered on a 40 µm Flowmi^®^ Cell Strainer, counted and cell viability was assessed (viability was 92% for the first experiment and 94% for the second). For each replicate, single-cell droplet libraries were generated from 16000 cells with the Chromium Single Cell 3′ GEM, Library & Gel Bead Kit v3 (10X Genomics, Pleasanton, CA, USA). After qPCR quantification, the libraries were pooled and sequenced on a single run, on a NovaSeq 6000 Sequencing System (Illumina), providing a read depth of >20,000 read pairs per cell, according to manufacturer’s instructions.

#### 2.4.2. Primary Analysis of Single-Cell Transcriptomic Data

The data were processed using cellranger 4.0.0 (10X Genomics). First, demultiplexing of the raw base call files into FASTQ files was accomplished using cellranger mkfastq function. Second, alignment on the GRCh38 reference genome, filtering and counting steps were performed, separately, on each replicate with the cellranger count function. Lastly, aggregation with the normalization of duplicates was performed using the cellranger aggr function.

#### 2.4.3. Secondary Analysis of Single-Cell Transcriptomic Data

The gene expression matrix was analyzed using the R package Seurat ([25]; RRID:SCR_016341). Doublets were identified and removed using the R package DoubletFinder ([26]; RRID:SCR_018771), assuming a 7.5% doublet formation rate. In addition, only cells with 200 to 5000 detected features and with <25% reads aligned to mitochondrial genes were selected for further analysis. After normalization, unwanted sources of intercellular variations, such as the number of detected genes or differences between cell cycle phases, were regressed using the ScaleData function. A principal component analysis was then performed using the 2000 most variable genes, according to the FindVariableFeatures function, and the first 10 components were used to calculate the UMAP. Cell-type labelling was performed using published single-cell RNA-seq data from a human fetal heart as a reference [4]. Cell-type labels from the reference were automatically transferred after cell-to-cell matching at the individual cell level using the R package CellID [27].

### 2.5. Musclemotion

hiPSC-CMs were filmed after glucose starvation at D30 in routine culture condition (37 °C, 5% CO_2_), without electrical stimulation, using Nikon A1 RSI confocal microscope with X20 Dry N.A 0.75 objective. MUSCLEMOTION software (v1.0; Gaussian Blur: No; Speed Window: 5; Noise Reduction: Yes; Automatic Reference Frame Detection: Yes; Transient analysis: Yes; [28]) was used to obtain contraction traces from 120 fps videos. Contraction profiles were analyzed using homemade R pipeline.

### 2.6. HEK293 Cell Culture and Transfection

The HEK293 cells were maintained at 37 °C, 5% CO_2_, in DMEM media with 10% FBS, 5% L-Glutamine and 5% Pen/Strep. The cells were plated in a 24-well plate or a 6-well plate and transfected the next day using FuGENE^®^ 6 (E2691, Promega, Madison, WI, USA). For the luciferase assay, the cells were transfected with a total of 2 µg of plasmid including: (1) pGL2-Renilla plasmid; (2) plasmid containing Firefly luciferase gene upstream promoter of interest; and (3) expression plasmids coding for proteins of interest (Table 1). The DNA quantities were equalized in each condition using empty pcDNA3.1 plasmid. The media was changed 24 h post-transfection, and cell lyses were performed 48 h post-transfection. For co-immunoprecipitation, the cells were transfected only with expression plasmids, prior to lysis, 24 h post-transfection.

### 2.7. Co-Immunoprecipitation

#### 2.7.1. Protein Sample Extraction and Quantification

Previously transfected HEK293 cells were lysed (4 °C, 15 min, with rotation) in lysis buffer: 1% TritonX-100, 100 mM NaCl, 50 mM Tris-HCl, 1 mM EGTA, 1 mM Na_3_VO_4_, 50 mM NaF, 1 mM phenylmethylsulfonyl fluoride, protease inhibitors cocktail (P8340, Sigma-Aldrich, Saint-Louis, MO, USA), and centrifuged at 15,000 g (4 °C, 15 min). Protein quantification was carried out using Pierce™ BCA Protein Assay Kit (Thermo Fisher Scientific, 23225).

#### 2.7.2. Bead-Antibody Complexes Preparation

Co-Immunoprecipitation was performed using Dynabeads^®^ Protein G (10004D, Invitrogen, Carlsbad, CA, USA) and DynaMag™-2 Magnet (Invitrogen, 12321D). First, 12.5 µL of beads were conjugated (Room temperature (RT), 40 min, with rotation) with 2 µg of antibody (Table 1). The bead-antibody complexes were cross-linked (RT, 30 min, with rotation) using 5.4 mg/mL dimethyl pimelimidate (21667, Thermo Fisher Scientific). The cross-linking was quenched with 50 mM Tris pH 7.5 (RT, 15 min, with rotation). Beads were washed using (1) PBS 1X; (2) 0.1 M citrate pH 3.1; (3) Na-phosphate solutions, then incubated in PBS 0.5% NaDOC (RT, 15 min, with rotation) and were finally washed with lysis buffer.

#### 2.7.3. Immunoprecipitation and Western Blotting Analysis

The bead-antibody complexes were incubated with 1 mg protein samples (4 °C, 2 h, with rotation). The supernatant was then discarded and the beads were washed 3 times with lysis buffer. The beads-protein complexes were then heated (50 °C, 10 min) in NuPAGE™ LDS Sample Buffer (4X) (Invitrogen, NP0008). The samples were magnetized prior supernatants collection and incubated (70 °C, 10 min) in NuPAGE^®^ Sample Reducing Agent 10X (Invitrogen, NP0009). Finally, the samples were loaded onto a 4–15% precast polyacrylamide gel (4568083, Biorad, Hercules, CA, USA) together with 10μg of total protein used as control. Revelation was performed using corresponding antibody (Table 1) with ECL Clarity Max (Biorad, 1705062). Images were acquired with ChemiDoc camera (Biorad) and analysed using Image Lab Software (Biorad).

### 2.8. Luciferase Assay

The cells were lysed according to the manufacturer recommendations and the luciferase activity was measured using Dual Luciferase reporter assay system (Promega, E1910) with Varioskan™ LUX microplate reader (Thermo Fisher Scientific). Mann-Whitney statistical tests were performed with Prism software (v8.0.1).

### 2.9. Immunofluorescence

The cells were fixed with 4% paraformaldehyde for 15 min at room temperature (RT) in Matrigel^®^-coated µ-Slide 8 Well (IBIDI, Gräfelfing, Germany) prior permeabilization with 0.1% PBS-BSA 1% Saponin (RT, 15 min) and blocking with 3% PBS-BSA (RT, 30 min). The cells were then incubated with primary antibodies (dilution 1/250) in PBS 0.1% BSA 0.1% Saponin solution (4 °C, overnight). Finally, the cells were washed and incubated with secondary antibodies and DAPI (RT, 1 h) and stored in 0.5% paraformaldehyde (4 °C). Images were acquired using an inverted epifluorescence microscope (Zeiss Axiovert 200 M).

### 2.10. TF and Cardiac Phenotypes Association

The association between cardiac phenotypes and transcription factors was performed using the DisGeNET (v7.0; [29]) and NHGRI-EBI GWAS Catalog [30] databases, filtering on cardiovascular traits, which were then manually validated.

### 2.11. Quantitative RT-PCR

The reverse transcription of 1 μg total RNA into cDNA was achieved using the high-capacity cDNA reverse transcription Kit (Thermo Fisher Scientific). Quantitative Polymerase Chain Reactions (qPCR) were executed in duplicates using TaqMan^®^ PCR Universal Master Mix (Thermo Fisher Scientific). TaqMan probes targeting GATA4 (Hs00171403_m1), IRX3 (Hs00735523_m1), IRX5 (Hs04334749_m1), and SCN5A (Hs00165693_m1). Threshold cycles (Cts) were normalized to ACTB (Hs99999903_m1).

## 3. Results

### 3.1. Directed Cardiac Differentiation Robustly Generates Functional Cardiac Cells

The cardiac differentiation of three hiPSC lines reprogrammed from three healthy donors was used as a cellular model of cardiac development (Figure 1A). After directed cardiac differentiation, all three hiPSC lines expressed cardiac-specific troponin I (Figure 1B), and displayed spontaneous contractions (Figure 1C), demonstrating their capability to form functional cardiomyocytes.

**Figure 1 cells-11-03915-f001:**
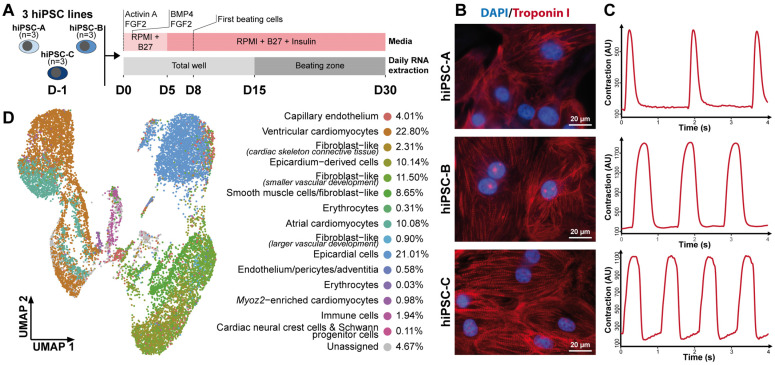
Transcriptomic and functional characterization of cardiac cells derived from hiPSCs. (**A**) Diagram illustrating the experimental design involving three distinct cardiac differentiations of three hiPSC lines reprogrammed from healthy donors. (**B**) Immunocytochemistry staining of troponin I (red) and DAPI (blue) at D30 of cardiac differentiation for all 3 hiPSC lines. (**C**) Representative contraction patterns captured by MUSCLEMOTION software on movies at D30 of cardiac differentiation for the 3 hiPSC lines. (**D**) UMAP displaying single-cell RNA-seq data at D30 of cardiac differentiation of hiPSC-A line. The color code indicates the different cell types identified. Cell population fractions are listed on the right. See also Appendix A.

Based on the single-cell transcriptomic data from 14,520 cells obtained at the end of directed cardiac differentiation (Figure 1D and Appendix A), approximately 95% of the cells could be successfully annotated to one of the 15 cell types described in the developing human fetal heart [4], including 34% cardiomyocytes, 21% epicardial cells, and 15% fibroblast-like cells. This distribution was similar to previous findings in the adult human heart [31]. These data indicate that directed cardiac hiPSC differentiation generated the cellular diversity observed in the human fetal heart, known to be not only necessary for cardiac function, but also required for the establishment of cardiomyocytes [32].

To investigate how gene expression variations are orchestrated throughout cardiac differentiation, we then generated daily transcriptomic data, from the hiPSC stage (D-1) to day 30 (D30), for three independent cardiac differentiations of each of the three hiPSC lines (Figure 2A). The directed cardiac differentiation was associated to the gradual temporal transcriptomic changes, represented on the first principal component (PC1) of the principal component analysis (Figure 2B). PC1 was significantly correlated with time from the onset of cardiac differentiation (spearman correlation coefficient rho = 0.87, *p*-value < 2.2 × 10^−16^). The cardiac differentiation evolution, represented by PC1 (Figure 2C), showed that 85% of the transcriptomic variations were achieved by D14 (Figure 2D). Altogether, these data demonstrate that the first 14 days of hiPSC cardiac differentiation represent the ideal time window to investigate the molecular processes that lead to functional cardiac cells.

### 3.2. Transcriptomic Kinetics of hiPSC Cardiac Differentiation Unveiled Biological Processes Involved during Cardiac Development

Focusing on gene expression changes related to hiPSC cardiac differentiation, the 3000 genes with the most significant expression variation during the directed cardiac differentiation (differentially expressed genes; DEG) were identified and grouped into 12 clusters, chronologically ordered based on the time point when their expression level changes the most, showing distinct temporal gene expression profiles (Figure 3A; Appendix A). The average temporal expression pattern of each cluster was then compared to the transcriptomic data obtained for the same genes from an in vivo reference model of murine cardiac development ([19]; Figure 3B). As cardiac cells derived from hiPSCs are usually described as reaching an equivalent of, at the most, E18.5 stage in murine embryonic development [33], we restricted the comparison of the hiPSC dataset to the murine developmental transcriptomic data obtained between murine embryonic stem cells and E18.5 stage. With the exception of cluster D, all of the clusters displayed strikingly similar expression patterns between hiPSC cardiac differentiation and murine cardiac development. Nevertheless, genes of cluster D were associated with gastrulation biological processes (Figure 3B—Cluster D middle panel) which is completed before E7.5. As no data were available between the mouse embryonic stem cell (mESC) and E7.5 stages in the murine experiments, the relevant gene expression changes associated to this process were likely to be absent in the murine transcriptomic dataset, but remained detectable in the daily hiPSCs cardiac differentiation dataset. For all other 11 clusters, hiPSC cardiac differentiation could be confidently matched to sequential gene expression waves that occur during murine cardiac development. Altogether, these clusters recapitulate key steps of cardiac development, including (1) expression decrease of genes related to pluripotency and stemness maintenance (Figure 3B—Cluster A to C), followed by the transient expression of related genes; (2) to gastrulation and mesoderm formation (Figure 3B—Cluster D); and (3) to early cardiac development (Figure 3B—Cluster E). These specific patterns were then followed by the successive implementation and persistence over time of gene expression waves that set up the sequential establishment of the functional cardiac phenotype (Figure 3B—Cluster F to L). To confirm these results, similar analyses were conducted on the top 3000 DEG during murine cardiac development from mESCs to E18.5 (Appendix A; Appendix A). This again revealed the consistency of the gene expression changes during hiPSC cardiac differentiation and during murine cardiac development. Collectively, these analyses demonstrate that hiPSC cardiac differentiation precisely recapitulates transcriptomic processes related to human and mouse cardiac development.

### 3.3. Prediction of Gene Regulatory Networks Governing hiPSC Cardiac Differentiation

TFs are known to be key players in developmental processes [6,34]. Therefore, to elucidate gene regulatory networks that underlie human cardiac development, gene expression analysis was then focused on all 216 TFs that were found to be differentially expressed during the time-course of cardiac differentiation (Figure 4A; Appendix A). Overall, 69% of these TFs have already been linked to cardiac (patho)physiological phenotypes (Appendix A). We chose to adapt an expression correlation score involving time delay (LEAP method, see Methods) to capture the gene associations that are hidden by time lags (i.e., time delay between the mRNA expression of the source gene and the mRNA expression of its target gene). Using this method on the 216 TFs, we predicted interactions that activated or inhibited the expression of target TFs by source TFs, building a regulatory network. This gene expression-based network included eleven 467 activating interactions and eleven 539 inhibitory interactions (Figure 4B left panel; Appendix A). We then evaluated the biological relevance of these TF interactions, using the STRING protein-protein interaction (PPI) database to generate an undirected PPI-based network restricted to the 216 TFs (Figure 4B right panel). Interestingly, 182 TFs (84%) were found to share at least one known PPI interaction. This included interactions between TFs belonging to the same gene cluster but, also, interactions between TFs from different gene clusters, suggesting a coordination between TFs to regulate the successive gene expression waves.

Comparing both networks, the gene expression-based network (LEAP-based) contained a greater amount of information than the PPI-based network (STRING-based) (Figure 4B). Indeed, although both networks were generated using the same TF query list, the density (i.e., normalized averaged number of neighbors) of the gene expression-based network was 5.5 fold higher than the PPI-based network. Deeper analysis showed that approximately 100% of the nodes and 80% of the links found in the PPI-based network were also found in the gene expression-based network (Figure 4C,D). Moreover, focusing on the links between successive expression clusters, more than 76% of those found in the PPI-based network were also found in the gene expression-based network (Figure 4E). Further confirming the accuracy of the gene expression-based strategy, sub-networks that have been well-described in the literature were also present in both networks: (1) the network composed of the main actors of pluripotency (e.g., POU5F1) and early phases of cardiac development (e.g., EOMES, MESP1; Figure 4F); and (2) the TF network implicated in cardiogenesis (e.g., ISL1, MEF2C; Figure 4G). This validated the relevance of such an approach to the expression correlation score, taking into account the time delay in comprehensively analyze TFs and their interactions throughout cardiac differentiation. Altogether, while the gene expression-based network confirmed already known and validated interactions, it also inferred 21,530 new interactions, unveiling numerous new hypotheses on TF networks that are potentially critical for cardiac development.

### 3.4. IRX3 and IRX5 Are Involved in Triggering Expression of GATA4, NKX2-5, TBX5 Cardiac Transcription Factor Network

Leveraging this new gene expression-based network to uncover new regulation mechanisms, and based on our previous focus of interest [7,10], we evaluated the IRX TF family’s involvement in the establishment of cardiac developmental processes. The expression levels of the six different IRX TF genes were analyzed during cardiac differentiation in the three hiPSC lines (Figure 5A). The expression of *IRX6* was undetectable and the expression of *IRX1* and *IRX2* did not vary over time. Only *IRX3, IRX4* and *IRX5* expression increased significantly between D-1 and D30 of cardiac differentiation. Interestingly, based on their expression profiles, *IRX3* and *IRX5* ranged from the earliest cardiac-specific gene cluster with an expression level that was maintained until the end of cardiac hiPSC differentiation (cluster F). This suggested a potential role for IRX3 and IRX5 in the early establishment of gene regulatory networks essential for cardiac fate, and beyond. In contrast, *IRX4* expression was detected in one of the latest clusters (cluster K). Therefore, we subsequently focused on both IRX3 and IRX5 TFs.

In order to investigate the role of IRX3 and IRX5 in cardiac differentiation progression, all their potential TF targets in the subsequent G to L clusters were extracted from the gene expression-based network (Figure 5B). Interestingly, this analysis revealed previously unknown regulatory links between cardiac TFs. Indeed, the master cardiac TF genes *GATA4* (cluster G), *NKX2-5* (cluster I) and *TBX5* (cluster L) were individually found to be potential targets of both IRX3 and IRX5. It is well established that GATA4 acts in a multiprotein complex with NKX2-5 (cluster I) and TBX5 (cluster L) cardiac TFs [6,35,36]. To further explore the potential new interactions, we then focused on the gene expression-based sub-network involving IRX3, IRX5, GATA4, NKX2-5 and TBX5, subsequently referred to as the IGNiTe sub-network (Figure 5C). In the IGNiTe sub-network, IRX3 and IRX5 were inferred as activators of *GATA4*, *NKX2-5* and *TBX5*, and confirming the literature [5,37,38,39], GATA4 was inferred as an activator of *NKX2-5* and both GATA4 and NKX2-5 were activators of *TBX5* expression.

In order to investigate the biological relevance of these inferred interactions, luciferase assays were conducted on GATA4, NKX2-5 and *TBX5* core promoters (Figure 5D). IRX3 and IRX5 proteins were, separately (fold changes 4.2 and 1.5 respectively) or together (fold change 4.5), able to bind the promoter of *GATA4* and to activate luciferase expression. A slight tendency towards the potentiation of both activating effects is observable when IRX3 and IRX5 were present, but this was not statistically significant. On the *NKX2-5* promoter, IRX5 alone was able to activate luciferase expression (1.3-fold change), but not IRX3, suggesting that the inferred activation of *NKX2-5* by IRX3 found in the IGNiTe sub-network was due to IRX5, and that the high similarity between the IRX3 and IRX5 expression profiles caused the false-positive link to appear. Together, IRX3 and IRX5 were able to activate the *NKX2-5* promoter, with a tendency towards potentiation (fold change 1.2 between IRX5 alone and IRX3/IRX5 conditions; *p* > 0.05). According to the order of appearance of the TFs in the IGNiTe sub-network, *NKX2-5* promoter activation was assessed in the combined presence of IRX3, IRX5 and GATA4, which showed an activator effect (1.8-fold change). Although a potentiation tendency was observed when GATA4 was present in addition to IRX3 and IRX5, this effect was not statistically significant. on the *TBX5* promoter, IRX3 and IRX5 were able to bind and activate gene expression either individually (2.7- and 1.2-fold change, respectively) or together (3.6-fold change). The potentiation of both activator effects was clearly observable and statistically significant when IRX3 and IRX5 were together on the *TBX5* promoter. Finally, considering the joint expression of IRX3, IRX5, GATA4 and NKX2-5 from D10, we proved the activator effect of these TFs on the *TBX5* promoter (6.6-fold change), which is statistically increased from the IRX3/IRX5 condition (1.8-fold change). Collectively, these results biologically validated the new interactions inferred with the gene expression-based network and illustrated the progressive temporal activation of the major TFs, GATA4, NKX2-5 and TBX5, by IRX3 and IRX5 during cardiac cell lineage establishment.

### 3.5. IRX3 and IRX5 Physically Interact with GATA4, NKX2-5 and TBX5 to Control SCN5A Expression

As the expression of the IGNiTe sub-network members was maintained until D30 of hiPSC cardiac differentiation (Figure 6A and Appendix A), the functional role of IRX3, IRX5, GATA4, NKX2-5, and TBX5 as a multiprotein complex was evaluated using co-immunoprecipitation and luciferase assays in heterologous expression systems. Luciferase assays were conducted on the promoter of *SCN5A*, a known target of these TFs [7,40,41,42,43]. According to the chronological order of expression of these five TFs along the cardiac differentiation of the hiPSCs (Figure 6A), we first investigated the role of IRX3 and IRX5. As previously described [44], IRX3 and IRX5 physically interacted (Figure 6B top panel and Appendix A) and could cooperatively activate the *SCN5A* promoter (Figure 6B bottom panel). While IRX3 alone activated the *SCN5A* promoter (2.3-fold change), IRX5 potentiated its effect with a 1.5-fold change. GATA4 was able to physically interact with IRX5 but not with IRX3 (Figure 6C top panels) and when the three TFs were co-transfected, only GATA4 and IRX5 interacted, suggesting a competitive effect between IRX3 and GATA4 to bind IRX5 (Figure 6C bottom left panel). Furthermore, the addition of GATA4 potentiated (1.5-fold change) the activity of the IRX3/IRX5 couple on *SCN5A* promoter (Figure 6C bottom right panel). NKX2-5 interacted with both IRX3 and IRX5 individually (Figure 6D left panels), but, again, when the four TFs were co-transfected we only observed an interaction between IRX5, GATA4, and NKX2-5, suggesting again a competition between IRX3 and IRX5, in favor of IRX5, in these interactions (Figure 6D central panel). NKX2-5 amplified (8.0-fold change) the effect of the IRX3/IRX5/GATA4 trio on the *SCN5A* promoter (Figure 6D right panels). Finally, when IRX3, IRX5, GATA4, NKX2-5 and TBX5 were co-transfected, a global protein complex could be formed between IRX5, GATA4, NKX2-5 and TBX5, but not with IRX3, even if IRX3 alone was able to interact with TBX5 (Figure 6E left and central panels). However here, TBX5 slightly reduced (−1.6-fold change) the effect of the IRX3/IRX5/GATA4/NKX2-5 quartet on the *SCN5A* promoter (Figure 6E right panel) suggesting a down-regulating role of TBX5 in this global complex. Collectively, we unveiled novel physical interactions between IRX TFs and three master cardiac TFs, GATA4, NKX2-5 and TBX5, that result in specific gene expression regulation. Furthermore, we showed that following IRX3, IRX5, GATA4, NKX2-5 and TBX5 gene expression increase during cardiac differentiation, the direct activation of *SCN5A* expression is under the control of a time-changing multi-TFs complex that controls the temporal expression profile of *SCN5A*.

## 4. Discussion

In this study, based on a transcriptomic kinetics study on the cardiac differentiation of hiPSCs, we identified the global TF regulatory network that is required for heart development. We notably identified novel time-dependent TF-gene regulations that connect IRX3 and IRX5 to the core cardiac GATA4, NKX2-5 and TBX5 TFs. We also found that these five TFs form protein complexes to regulate target gene expression, such as *SCN5A*. Altogether, this time-course bulk transcriptomic data provided a dynamic model relevant for identifying new roles for TFs in developmental processes.

### 4.1. In Vitro Modeling of Time in Cardiac Development

This study demonstrates that hiPSC cardiac differentiation is a relevant model to study the successive steps leading to the establishment of the gene expression program during human cardiac development. To date, most studies contributing to the knowledge on heart development and TF regulation have been conducted in animal models, primarily in mice [45], as access to human embryonic cardiac tissue is indeed very limited. If regulatory mechanisms of development are overall highly evolutionary conserved, some are human-specific [46,47]. Therefore, the investigation of human cardiac development also requires suitable human models. HiPSC cardiac differentiation models have proven to generate functional cardiac cells and suggested that punctual time points during this differentiation might reflect some key developmental stages [7,48,49]. However, entirely assessing the relevance of the hiPSC cardiac differentiation model for studying human cardiac development requires demonstrating that it thoroughly and accurately reproduces human cardiac development in a temporally coordinated fashion. All phenotypic changes that occur during cardiogenesis are known to be embodied by dynamic alterations in cellular transcriptome. Yet, although the ideal situation would be to compare transcriptomic changes along hiPSC differentiation to the ones occurring during human cardiac development, no public human transcriptomic dataset studying well-distributed stages across the entire cardiac development is available. In the present study, we therefore used murine cardiac transcriptomic data generated from specific stages that appropriately ranged from conception to birth [19], to compare with hiPSC cardiac differentiation data. Their high level of consistency confirmed that our hiPSC cardiac differentiation model accurately reproduces cardiogenesis. An important added value of the present data is that it filled a gap of knowledge on the global gene expression changes that occur daily between these developmental stages in human cells.

A major limitation of the hiPSC-derived models is its immaturity: cardiac cells produced by current hiPSC differentiation protocols have a fetal-like phenotype that is far from adult cells [50]. Although this limitation does not affect the study of pre-natal stages of cardiac development, obtaining mature cardiac cells would broaden the scope of these models to study later stages of development as well as aging processes.

### 4.2. In Vitro Modeling of Cardiac Development-Associated Cellular Diversity

Cardiomyocytes require substantial cell diversity to support both the proper execution of their biological functions and their differentiation, as many signaling pathways regulating their formation are sourced from other cell types [31,32]. In this study, we confirmed that hiPSC cardiac differentiation generates the cellular diversity typically reported in the human fetal heart and thus provides the opportunity to investigate regulatory mechanisms occurring between these different cardiac cell types. However, hiPSC cardiac differentiation in two dimensions does not reproduce the spatial organization of the cell types observed in the context of the heart. The emergence of more integrated hiPSC-derived models, such as cardioids [51], will therefore undoubtedly enhance our insights into transcriptional regulation between cardiac cell types.

### 4.3. Uncovering New Regulatory Networks Using a Gene Expression Kinetics-Based Strategy

An original aspect of this study was the identification of expression regulations occurring between TFs in a temporal manner. For this, we chose to adapt the LEAP bioinformatic tool designed for single-cell data to kinetic transcriptomic bulk data [52]. Importantly, with this tool, these gene regulations are oriented, indicating not only the interaction but also which partner is expected to be the target/source. This higher level of information is important for designing a more efficient confirmation experiment, and cannot be obtained in traditionally used protein-protein interaction databases, such as STRING [23]. Moreover, our strategy allowed us to biologically link genes in a time-dependent manner during cardiac differentiation, and thus provided important new insights on the cardiac gene regulatory networks [53]. Of note, one cannot exclude that some of the inferred links may not reflect biological interactions (e.g., TF does not directly bind to an inferred target gene). Other studies embarked in different strategies to study cardiac gene regulation. For instance, Gonzalez-Teran et al. combined PPI data associated with GATA4 and TBX5 TFs and genetic data generated on patients presenting congenital heart diseases (CHD) to identify CHD candidate genes [54]. This integrated strategy of PPI data and CHD-associated genetic data could be a relevant complementary approach to our chronological gene expression-based strategy in order to identify new CHD-associated TF regulatory networks and to offer a better understanding of the underlying mechanisms of cardiac diseases.

### 4.4. Activation Cascade of GATA4, NKX2-5, TBX5 Genes Triggered by IRX3 and IRX5

It is well established that cardiac transcription factors regulate the expression of other TF-coding genes. For instance, GATA4 activates *NKX2-5* expression and both GATA4 and NKX2-5 activates *TBX5* expression [5,37,38,39]. However, the precise molecular bases of these regulations were still to be uncovered. Using daily-generated transcriptomic data, we characterized the course of expression of these major cardiac TFs showing that, in accordance with the functional data, they are successively launched, starting with *GATA4* around day five, followed by *NKX2-5* two days later and finally by *TBX5* two days later. This raised the question of how *GATA4* expression is, in the first place, launched. Using the gene expression-based network, we identified IRX3 and IRX5 TFs as potential activators of *GATA4* expression. Accordingly, the expression of these TFs was launched simultaneously about one day prior to *GATA4* expression. These TFs are of growing interest as, while most studies were performed in knockout mice showing that they play redundant roles in cardiac development leading to embryonic lethality and in postnatal electrophysiological function, their role in human cardiac function now also emerges [7,55]. In this context, the present study therefore further explored and specified the role of IRX TFs in the course of human cardiac development.

### 4.5. Exploring the Functional Interplay between IRX3/IRX5 and GATA4, NKX2-5, TBX5

It is has been shown that GATA4, NKX2-5 and TBX5 act as multiprotein complexes to regulate cardiac gene expression [37]. Here, we completed this knowledge by showing that IRX3 and IRX5 can also physically bind to this TF regulatory complex. Furthermore, all five TFs could physically and functionally interact on the promoter of *SCN5A* that encodes the major cardiac sodium ion channel. Accordingly, *SCN5A* expression gradually increases during hiPSC cardiac differentiation, paralleling the progressive expression establishment of the five TFs. Some of the interactions between IRX TFs and GATA4, NKX2-5 or TBX5 have previously been published. For instance, physical and functional interactions between Irx3, Nkx2-5 and Tbx5 have been shown in mice to regulate genes implicated in ventricular conduction system establishment and maturation [56]. Furthermore, our group has previously demonstrated physical and functional interactions between IRX5 and GATA4 on *SCN5A* promoter [7]. In this study we further detailed the complexity of the interactions between IRX3, IRX5, GATA4, NKX2-5 and TBX5, and how these TF complex compositions impact the expression of a target gene. In a future work, it will be of great interest to investigate more precisely how these TFs directly and/or indirectly bind to gene promoters during cardiac differentiation of hiPSCs. For that, generating data that combine high throughput identification of TF binding sites together with ChIP-seq data for these TFs at specific time points of the differentiation protocol, would be central.

### 4.6. Perspectives

Altogether, this study provides a comprehensive dynamic blueprint of transcription factors that control transcriptional regulation during human cardiac development as well as a new methodological approach that may be applied to other research fields. These insights may help to further understand both pathological cardiac development leading to CHD, as well as physiological cardiac development, which is a prerequisite to emerging cardiac regenerative therapy strategies [57]. Moreover, in recent years, transcription factor regulation of cardiac functions has been widely supported by Genome Wide Association Studies, linking numerous common genetic variations at loci harboring TF genes to cardiac diseases ([58,59]; Appendix A). Confronting the present knowledge with that obtained from the cardiac differentiation of hiPSCs reprogrammed from patients carrying such genetic variants may provide important information regarding their impact on cardiac development and therefore may lead to new targets for treatment and clinical management improvement.

### 4.7. Limitations of the Study

We have identified several limitations in our study. First, the kinetic transcriptomic analysis has been performed using bulk-based strategy; however, the use of single-cell analysis instead would have provided us with a better overview of the cellular transcriptomic diversity. Second, the limited sample size that has been used prevented us from identifying the impact of the gender and ethnicity on transcriptomic regulation. Therefore, further studies will have to investigate if the identified TF networks (1) are activated in a cell-specific manner; and (2) whether they are specific to gender and/or ethnicity.

## Figures and Tables

**Figure 2 cells-11-03915-f002:**
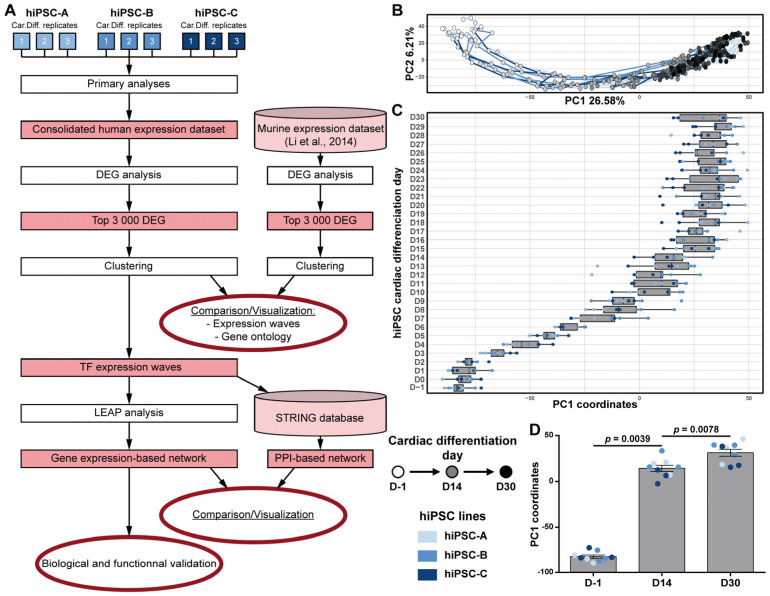
Transcriptomic time-course analysis of hiPSC cardiac differentiation. (**A**) Methodological workflow. Steps are represented in white rectangle and outputs in red rectangles. (**B**) Global transcriptomic variations displayed with the first two components of the Principal Component Analysis. Three cardiac differentiations were studied for each of the three hiPSC lines. For each cardiac differentiation, a line connects the time-points in chronological order. (**C**) Boxplots displaying the distribution of PC1 coordinates of each replicates at each day (median ± quartile). (**D**) Histogram comparing distribution of PC1 coordinates at the beginning (D-1), the middle (D14) and the end (D30) of hiPSC cardiac differentiations (Mean ± SEM; Wilcoxon matched-pairs signed rank test).

**Figure 3 cells-11-03915-f003:**
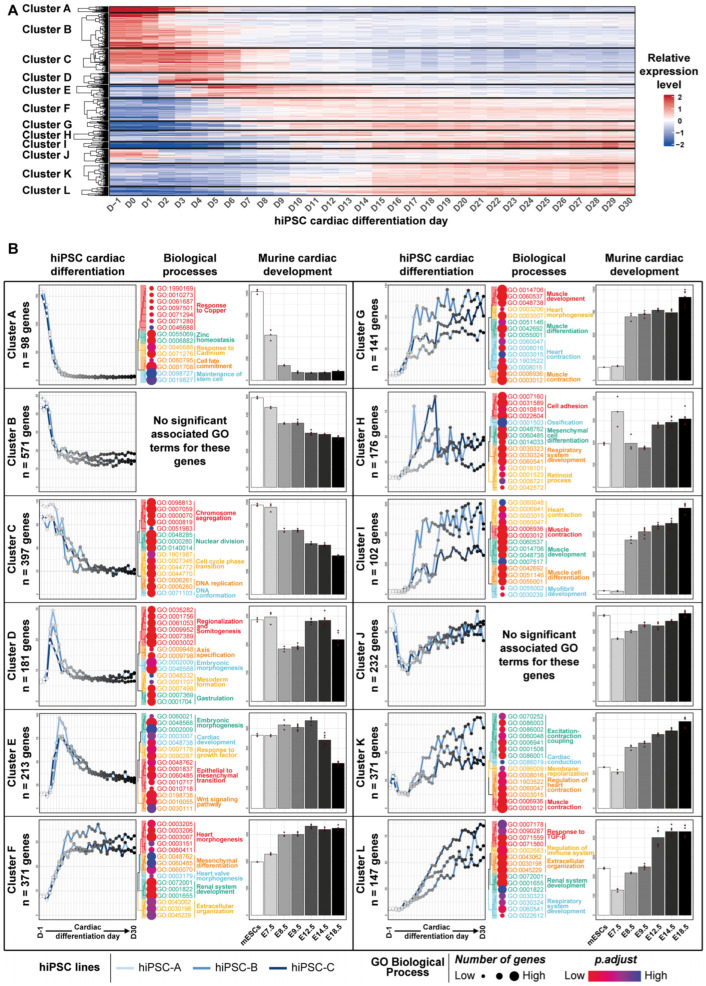
Expression profile and functional annotation of the top 3000 differentially expressed genes (DEGs) during hiPSC cardiac differentiation, and comparison with murine cardiac development gene expression dataset. (**A**) Heatmap displaying DEG expression levels. The entire data set was used to aggregate the genes into 12 clusters and the mean expression level of 9 replicates is represented. (**B**) For each cluster, average gene expression level during hiPSC cardiac differentiation (left panel for each cluster) and of their orthologs during murine cardiac development (mESCs to E18.5 stage, right panel for each cluster) are shown. Replicate gene expression levels were averaged for each hiPSC line (n = 3 per hiPSC line and per timepoint) and for murine data (n = 3 to n = 6 per timepoint, depending on the stage). The 15 most significantly related GO terms are displayed for each cluster on the middle panel. See also Appendix A.

**Figure 4 cells-11-03915-f004:**
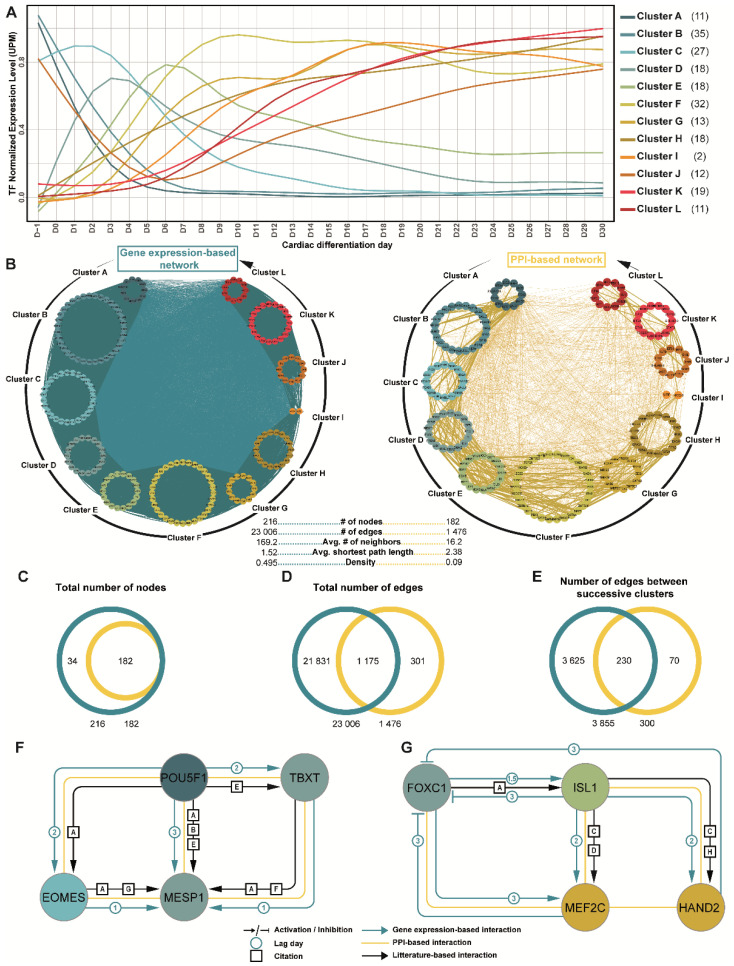
Inferred TF regulatory network governing hiPSC cardiac differentiation. (**A**) Normalized gene expression of the 216 TFs (identified within the top 3000 differentially expressed genes during hiPSC cardiac differentiation) were quantified and averaged in each gene cluster. UPM: UMI per million. Numbers in brackets indicates the TF number per cluster. (**B**) Graphical representation of gene expression-based network and protein-protein interaction-based network (LEAP- and STRING- based method, respectively) of the same TFs as in (**A**). Interactions between TFs of successive clusters are shown using bold lines. #: quantity of each analyzed parameter. (**C**–**E**) Comparative quantitative analysis between both networks. (**F**,**G**) Examples of two literature-based sub-networks. Interactions uncovered in gene expression-based network are shown in blue, in PPI-based network, in yellow, and by literature curation, in black. Node colors correspond to the one of their corresponding gene cluster (as in (**A**)). Paper DOIs associated with literature-based links: [A] 10.3390/genes12030390; [B] 10.1002/stem.1362; [C] 10.3389/fcell.2021.793605; [D] 10.1242/dev.01256; [E] 10.1016/j.devcel.2006.07.013; [F] 10.1093/cvr/cvr158; [G] 10.1038/embor.2012.23; [H] 10.1242/dev.073056.

**Figure 5 cells-11-03915-f005:**
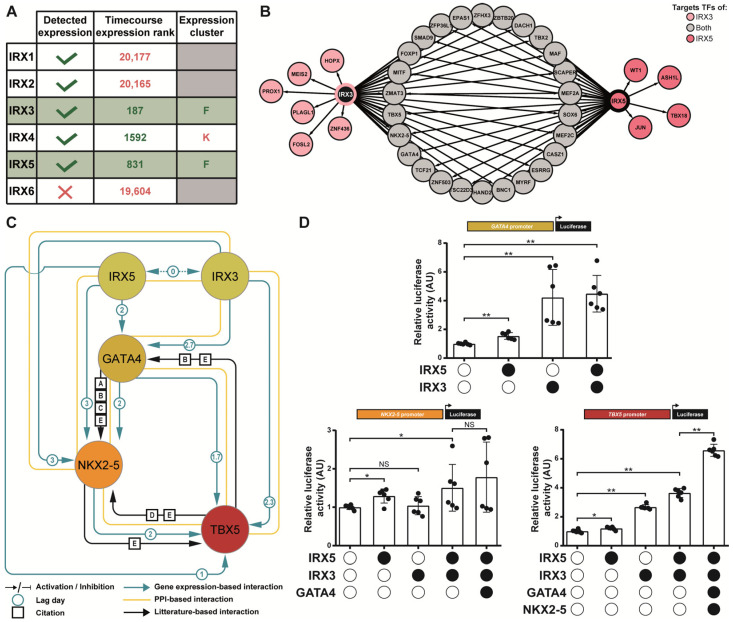
Exploration of the inferred IIGNT sub-network. (**A**) Table of TF selection criteria for the IRX genes family. Timecourse expression rank is the output of the timecourse package and illustrates the variation in gene expression during directed cardiac differentiation (a lower number indicating a higher variation). The Expression cluster column refers at the expression cluster in which each TF ranged as in Figure 4A. Selection and exclusion criteria are indicated in green and red respectively. (**B**) Potential target TFs of IRX3 and/or IRX5 identified in the G-to-L clusters based on the gene expression-based network. (**C**) Gene expression-based network of IRX3, IRX5, GATA4, NKX2-5 and TBX5 TFs. Node colors represent their corresponding clusters as in Figure 4A: IRX3 and IRX5—cluster F; GATA4—cluster G; NKX2-5—cluster I; TBX5—cluster L. Lag is shown in days. References to literature-based links: [A] 10.1016/B978-0-12-381332-9.00027-X.; [B] 10.1101/cshperspect.a008292; [C] 10.1016/B978-0-12-387786-4.00008-7; [D] 10.1016/j.mod.2020.103615; [E] 10.1101/cshperspect.a013839. (**D**) Graphs displaying activity levels of luciferase that is under the control of *GATA4* (-1800_TSS_+200), *NKX2-5* (-2000 bp_Start codon) and *TBX5* (-1800_TSS_+200) promoter constructs. Mean ± SD; * and **: *p* < 0.05 and *p* < 0.01, respectively (Mann-Whitney test).

**Figure 6 cells-11-03915-f006:**
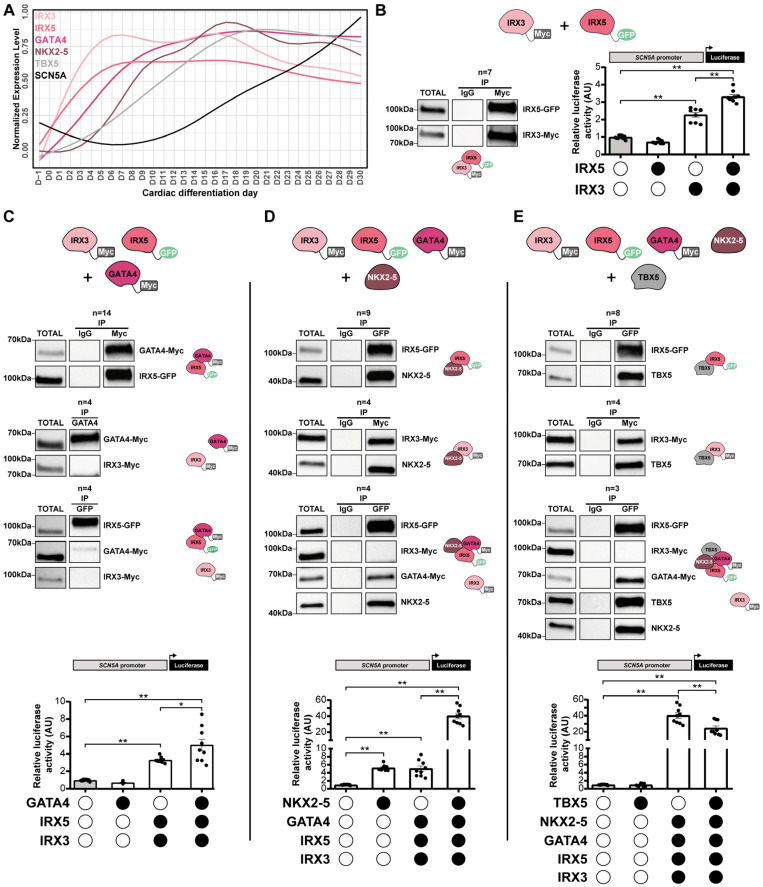
Physical and functional interactions of IRX3/IRX5/GATA4/NKX2-5/TBX5 multiprotein complex. (**A**) Normalized mean expression level overtime of IRX3, IRX5, GATA4, NKX2-5, TBX5 and of *SCN5A* ion channel genes. UPM: UMI per million. (**B**–**E**) Co-immunoprecipitation and luciferase results associated to the transfection of (**B**) IRX3 and/or IRX5, (**C**) IRX3, IRX5 and/or GATA4, (**D**) IRX3, IRX5, GATA4 and/or NKX2-5, (**E**) IRX3, IRX5, GATA4, NKX2-5 and/or TBX5. Immunoblots representative of the various co-immunoprecipitations and the schematic illustration of the results. Graphs display activity levels of luciferase that is under the control of the −2109/+1072 region of human *SCN5A* promoter, in the various transfection conditions. Mean ± SEM; * and **: *p* < 0.05 and *p* < 0.0001, respectively (Mann-Whitney test). See also Appendix A.

**Table 1 cells-11-03915-t001:** Plasmids and antibodies references.

Plasmid Name	Sequence/Reference	Supplier	
*NKX2.5* promoter–FireflyLuc	-2000 bp_Start codon	Vectorbuilder	
*GATA4* promoter–FireflyLuc	-1800_TSS_+200	Vectorbuilder	
*TBX5* promoter–FireflyLuc	-1800_TSS_+200	Vectorbuilder	
*SCN5A* promoter–FireflyLuc	-2109_TSS_+1072	Adapted from [7]	
pGL2 Renilla luciferase		Promega	
*IRX5*	RG234228	Origene	
*IRX3*	RG205722	Origene	
*GATA4*	RC210945	Origene	
*TBX5*	SC120046	Origene	
*NKX2.5*	SC122678	Origene	
pcDNA3.1		Invitrogen	
**Antibody**	**Reference**	**RRID**	**Supplier**
anti-GFP	TA150041	AB_2622256	Origene
anti-Myc Tag	05-724	AB_309938	Merck Millipore
anti-IRX5	sc-81102	AB_1124818	Santa Cruz
anti-IRX3	sc-166877	AB_10609525	Santa Cruz
anti-GATA4	sc-25310	AB_627667	Santa Cruz
anti-TBX5	sc-515536		Santa Cruz
anti-NKX2.5	sc-8697	AB_650280	Santa Cruz
anti-Troponin I	sc-15368	AB_793465	Santa Cruz
Mouse IgG Isotype Control	02-6502	AB_2532951	Thermo Fisher Scientific

## Data Availability

The authors declare that all data supporting the findings of this study are available within the article and its supplementary information files. Bulk and single-cell transcriptomic data are available in ArrayExpress database at EMBL-EBI (https://www.ebi.ac.uk/arrayexpress) under accession number E-MTAB-11822 and E-MTAB-11817, respectively. All original codes have been deposited at Gitlab (https://gitlab.univ-nantes.fr/E132534J/cardiff.git). Any additional information required to reanalyze the data reported in this work is available from the lead contact upon reasonable request.

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
