# Peer review of "Deciphering Transcriptional Networks during Human Cardiac Development"

_cells, 2022, doi:10.3390/cells11233915_

Round 1
Reviewer 1 Report
Very strong comprehensive study from both experimental and bioinformatics/systems biology points of view. I have only minor concerns to authors to try to make pictures more legible (especially 3B and 4B).
Reviewer 2 Report
In this manuscript, Canac, Cimarosti and colleagues describe an extensive experimental and computational analysis aimed at the identification of a set of key components of part of the transcriptional network activated during human cardiac development.
The authors used an integrated omics approach, applied to a biological model of hiPSC cells upon cardiac differentiation comprising a single-cell transcriptomic profiling (Figure 1), together with a bulk transcriptomic time-course analysis (Figure 2 and 3). The results of a bioinformatics analysis highlighted a set of TFs that were further studied in terms of their regulatory relationships (Figure 4). Exploration of the inferred sub-network involving IRX3, IRX5, GATA4, NKX2-5, and TBX5 genes generates as a result the hypothesis of a physical interaction among them (Figure 5 and 6).
In my opinion, this is an interesting work. The major point here is the achievement of a wide set of omics experiments that allowed the production of a set a large set of data associated with the study of human cardiac development.
I have the following comments:
1) It is not clear to me the meaning of the regulatory interactions described in particular in the second part of the work. Do the TFs actually bind the promoter of the target genes ? This is somewhat mentioned in the discussion section, however this is an important point that must be better analyzed. For example, are TFBSs (TF binding sites) for the considered TFs actually present on the promoter regions of potentially regulated genes ?Can the author make some investigations on that ?
2) Another point that needs further clarifications is the actual meaning of the complex between the GATA4, NKX2-5, TBX5, IRX3 and IRX5 genes. If I understand correctly, some of the proposed interactions are already known. A comment could help the readers in focusing on the original result of this paper in term of the regulation of SCN5A gene.
3) Can the author validate some of the genes reported in the text at a single-gene level, like with RT-PCR ?
4) Some data must be added. I would like to see an Excel file reporting data generating heatmap of Figure 3A, with the explicit association between genes and clusters; plus another one reporting the original lists of the top 3000 genes DEGs used for the analysis, for both human and mouse.
